# Foundation models enable wearable signal screening for cardiovascular disease among people living with HIV
Munib Mesinovic [1] ✉, Hai Ho Bich [2], Ly Vo Trieu[3], Viet Nguyen Quoc[3], Ngoc Nguyen Thanh[2], Tuan Anh Nguyen Hoang[4], Minh Tu Van Hoang[2], Phan Nguyen Quoc Khanh [2], Xuan Huy Vo[3], Phuc Vo Hong [2], Khoa Le Dinh Van[2], Yen Lam Minh[2], Louise Thwaites [2] & Tingting Zhu [1]

## Abstract

**Background** Cardiovascular disease screening faces significant challenges in resource-limited settings, where infrastructure and computational constraints preclude advanced assessment. These constraints are particularly acute for people living with human immunodeficiency virus (HIV), who experience elevated cardiovascular risk yet often receive care in clinics without specialist diagnostic capacity. Pretrained physiological foundation models offer potential for low-cost screening using wearable sensors, though their applicability in resource-constrained settings remains unclear.
**Methods** We evaluate pretrained physiological embeddings from foundation models for cardiovascular disease detection using photoplethysmography signals from 80 people living with HIV in Ho Chi Minh City, Vietnam. Of 80 participants, 13 (16%) had cardiologist-confirmed cardiovascular disease. We compare strictly zero-shot deployment (NormWear without local training) with frozen PaPaGei embeddings plus locally trained classifier, alongside traditional approaches.
**Results** Here we show that the PaPaGei-embedding approach achieves area under the receiver operating characteristic curve 0.769 (95% confidence interval: 0.70, 0.84) and average precision 0.489 (0.37, 0.61) in this pilot cohort, numerically higher than zero-shot NormWear (0.610; 0.226), principal component analysis features (0.651; 0.208), and supervised clinical models (0.744; 0.433). This approach requires local labels for classifier training but avoids computationally intensive foundation model fine-tuning. However, given the small positive class size (13 cases), these findings require validation in larger cohorts. PaPaGei embeddings capture clinically coherent structure: patients on dolutegravir-based regimens cluster in low-risk regions, while those with high cholesterol variability occupy high-risk areas.
**Conclusions** These preliminary findings provide a potential methodological framework for deploying foundation models in resource-constrained settings, though adequately powered, multi-centre validation is essential before clinical implementation.

## Plain language summary

People living with human immunodeficiency virus (HIV) face higher risks of heart disease, yet many clinics in areas most vulnerable lack expensive equipment for heart screening. We tested whether artificial intelligence combined with low-cost wearable pulse sensors could help identify heart problems in 80 people living with HIV in Vietnam. Thirteen participants had confirmed heart disease based on specialist assessment. We found that using pretrained artificial intelligence models with these simple sensors successfully identified most heart problems without requiring powerful computers or extensive training data. The mathematical predictive model predicted high risk of heart disease among participants with known risk factors, such as changes in cholesterol. While these early findings need testing in larger groups, this approach could help clinics with limited resources screen patients for heart disease using affordable technology instead of expensive specialist equipment.

Cardiovascular disease (CVD) screening is challenging to deliver at scale in resource-limited settings, where laboratory capacity, specialist diagnostics, and advanced computational infrastructure are often limited[1–3]. These structural barriers limit both traditional clinic-based approaches and data-hungry machine-learning pipelines that depend on large, annotated local datasets[4]. Practical, label-efficient strategies are therefore needed to enable robust triage within routine outpatient workflows. The challenge is particularly acute for people living with HIV (PLWH), who face a 1.5–2-fold

[1]Department of Engineering Science, University of Oxford, Oxford, UK. [2]Oxford University Clinical Research Unit, Ho Chi Minh City, Vietnam. [3]Hospital for Tropical Diseases, Ho Chi Minh City, Vietnam. [4]People's Hospital 115, Ho Chi Minh City, Vietnam. ✉e-mail: munib.mesinovic@eng.ox.ac.uk

higher risk of CVD than the general population[5–7]. In Vietnam, CVD accounts for 31% of all mortality, yet screening infrastructure remains limited and specialist services are concentrated in urban centres[8,9]. These constraints hamper early identification of contemporaneous cardiac abnormalities among PLWH and complicate efforts to deploy conventional CVD detection tools or fully supervised models.

Despite the pressing need for scalable CVD screening in PLWH, machine learning approaches remain significantly underutilised in routine clinical practice for this population. A recent systematic review found that the vast majority of cardiovascular models for PLWH still rely on traditional statistical and simplistic linear approaches[10]. The limited adoption stems from multiple barriers: most ML models require extensive labelled datasets unavailable in resource-constrained settings[11], lack interpretability needed for clinical trust[12], and have not been validated in PLWH populations[13]. Furthermore, existing ML applications in cardiovascular medicine have predominantly focused on high-resource settings with advanced imaging modalities[14], leaving a critical gap for accessible, wearable-based approaches suitable for outpatient HIV care.

Photoplethysmography (PPG), obtainable via low-cost wearables, is an attractive modality for scalable, remote assessment[15,16]. Heart rate variability (HRV) and waveform morphology features derived from PPG correlate with cardiovascular status and adverse events[17–19]. Pipelines based on handcrafted features and supervised classifiers, however, require careful preprocessing and disease-specific training data, limiting their generalisability across populations and care settings[20–23]. In PLWH specifically, established risk scores and conventional ML approaches have systematically underpredicted risk[10].

Foundation models trained on large-scale physiological data offer a practical alternative by providing rich pretrained representations that can be used with minimal local supervision. Models such as NormWear[24] and PaPaGei[25] extract embeddings from ECG and PPG that capture complex physiological structure. While these encoders can be applied in a strictly zero-shot manner, real-world deployment in low-resource clinics often benefits from lightweight local calibration, training only a simple classifier on frozen embeddings rather than fine-tuning the encoder, thereby balancing performance with computational feasibility, though this still requires local labelled data.

Here, we evaluate pretrained physiological embeddings for CVD screening among 80 PLWH outpatients attending the clinic for routine medication collection in Ho Chi Minh City, Vietnam. We compare deployment modes ranging from strictly zero-shot application to a label-efficient pipeline using frozen PPG embeddings from PaPaGei with a locally calibrated classifier. We assess each encoder in its native modality (PaPaGei on PPG; NormWear on ECG/PPG for zero-shot baselines) to avoid cross-domain artefacts. In this proof-of-concept pilot study, PaPaGei embeddings with a lightweight classifier achieve numerically higher discrimination (AUROC 0.769) over zero-shot and clinical-score comparators in our cohort, while aligning with known risk factors. However, given the small sample size ($N = 80$, 13 CVD cases), these findings require validation in larger studies before any clinical deployment. This work provides a potential methodological framework for deploying foundation models in resource-constrained settings to support equitable, scalable CVD prescreening for PLWH, pending rigorous external validation.

## Methods

### Data acquisition and processing

This study was conducted at the Hospital for Tropical Diseases in Ho Chi Minh City, Vietnam, between 17 November 2023 and 20 July 2024. The protocol was approved by the institutional ethics committee and the Oxford Tropical Research Ethics Committee, with all participants providing written informed consent. Eligible individuals were adults (≥18 years) receiving routine outpatient HIV care with no acute illness or indication for hospital admission.

At baseline, each participant completed a standardised assessment including demographic variables (age, sex, year of HIV diagnosis, smoking history), antiretroviral therapy (ART) regimen, medication use, systolic and diastolic blood pressure, and anthropometric measures. Laboratory tests included total cholesterol, HDL cholesterol, CD4 count, and HIV viral load. Cardiovascular risk was assessed using the Framingham Risk Score for general cardiovascular disease[26] and its D:A:D-modified version, each calculated using baseline clinical and laboratory inputs. These risk scores were used to characterise baseline cardiovascular risk in the study population, not as direct performance comparators for our diagnostic approach, as they predict future cardiovascular events (5–10 year risk) rather than detect current abnormalities. Participants also underwent 12-lead electrocardiography (ECG), transthoracic echocardiography, and wearable photoplethysmography (PPG) monitoring using a SmartCare pulse oximeter (SmartCare Analytics UK). ECG and echocardiogram interpretations were independently reviewed by two cardiologists, with CVD-positive status assigned if either modality showed clinically significant abnormalities warranting further investigation or treatment.

Standard electrocardiography (ECG) was performed using Shimmer wearable ECG sensors, recording three leads: ECG LA-RA (Lead I), ECG LL-RA (Lead II), and ECG Vx-RL (chest lead Vx) in 24-bit resolution. ECG data were acquired as tab-separated files with calibration flags, with multiple recordings per visit concatenated after quality filtering. Raw ECG signals from the three leads were loaded and passed directly to the NormWear encoder for embedding extraction. NormWear produced patient-level embeddings by averaging segment-level representations. When concatenating with clinical features, these embeddings were then reduced to 15 dimensions using principal component analysis (PCA). No fine-tuning or retraining of the NormWear encoder was performed as the model was applied in its pretrained state.

For PPG signal processing, two distinct preprocessing pipelines were employed adapting to each foundation model's setup. For NormWear PPG embeddings (zero-shot baseline), raw PPG signals from the PLETH channel were segmented into overlapping windows of 128 samples with a stride of 64 samples. No resampling, z-score normalisation, or extensive filtering was applied beyond basic quality control, representing a minimally processed, computationally constrained deployment scenario. Windows were passed directly to NormWear in its pretrained state without local training.

For patients with recordings across multiple follow-up visits (up to 4 visits over 1 year), physiological signals from all available visits were pooled together. Each visit's PPG recording was independently segmented into 10 s windows, quality-filtered, and processed through the PaPaGei foundation model. Similarly, each visit's ECG recording was segmented into overlapping windows (128 samples, stride 64 samples) and processed through NormWear. The resulting embeddings from all windows across all visits were mean-pooled at the patient level to produce a single patient-level representation. This approach treats each visit as providing additional physiological samples, capturing both within-visit and between-visit variability whilst maximising available data. However, we note that this strategy does not explicitly model temporal dynamics or temporal ordering between visits; embeddings are aggregated irrespective of visit sequence. The binary CVD outcome reflects any evidence of cardiovascular abnormalities detected during the 1-year follow-up period based on ECG or echocardiographic findings at any visit.

For PaPaGei embeddings, PPG signals underwent model-specific preprocessing optimised for PaPaGei's input requirements. Raw PLETH signals sampled at 64 Hz were first normalised using z-score standardisation, followed by denoising and detrending using the `pre-process_one_ppg_signal` function from the preprocessing pipeline. Signals were then segmented into 10 s windows at the original sampling rate (640 samples per window), resampled to 125 Hz using signal interpolation to match PaPaGei's expected input frequency, and padded or truncated to exactly 1250 samples (10 s × 125 Hz). Padding was applied symmetrically when segments were shorter than 1250 samples. This preprocessing follows PaPaGei's documented specifications for optimal embedding quality. Processed segments were passed through the frozen PaPaGei encoder via prompt-based inference to generate fixed-length physiological embeddings,

**Fig. 1 | End-to-end embedding-based screening pipeline using PaPaGei.** PPG signals were recorded from PLWH using low-cost wearable pulse oximeters (SmartCare) during clinic visits. Signals were segmented into 10-s windows, quality filtered, and normalised before being passed through the PaPaGei foundation model using structured prompting. The resulting physiological embeddings were aggregated at the patient level and passed to a calibrated Random Forest classifier. This pipeline achieved the highest predictive performance across all tested configurations (AUROC = 0.769, AP = 0.489, recall = 1.000), outperforming traditional clinical risk scores and supervised baselines. The approach requires no encoder fine-tuning and no laboratory data, and uses a lightweight local classifier for calibration, making it viable for scalable prescreening in outpatient workflows.

which were then mean-pooled across all segments per patient and reduced to 15 dimensions via PCA before being passed to a calibrated Random Forest classifier.

Follow-up visits occurred every 3 months and included repeat ECG, PPG, and cardiovascular risk scoring. PPG signals were acquired using the SmartCare BM2000A wearable pulse oximeter following a standardised protocol (see Supplementary Section 1.2 and Fig. S1 for device setup details). The non-invasive monitoring setup enabled data collection during routine clinic visits without specialised infrastructure. PPG waveforms were recorded at 100 Hz via Bluetooth transmission to a mobile application (SmartCare Capture, available on Google Play Store). Recordings lasted ~20 min, with the first 15 min used for analysis. Raw PPG signals underwent standardised preprocessing and quality control using the Vital_SQI package[27], including noise filtering and segment rejection. Handcrafted heart rate variability (HRV) and waveform morphology features were extracted using the Vital_DSP[28] library. HRV features were summarised across all segments per participant by computing the mean, standard deviation, minimum, and maximum of each feature.

Clinical features were derived from a combination of single-timepoint variables (Visit 1) and multi-visit time series. Time-varying features, systolic/diastolic blood pressure and cholesterol, were collected across four outpatient visits and imputed row-wise using each patient's mean value where missing. Temporal statistics (mean, standard deviation, min, max, range, slope, and delta between Visit 1 and Visit 4) were computed per feature to capture dynamic risk patterns. ART regimen variables, smoking status, and cardiovascular history were retained as binary indicators. Derived features such as years on ART (year of treatment minus diagnosis) and categorical medication usage were also included. The Framingham and D:A:D-modified Framingham risk scores were discretised to binary indicators for comparison. For Framingham, moderate and high risk categories (score = 2) were mapped to 1, and low risk to 0. For D:A:D, we mapped high and very high risk categories (scores 2 and 3) to 1.

All supervised machine learning models were implemented in Python using scikit-learn and https://github.com/PriorLabs/TabPFN. We trained Random Forest, ElasticNet, LightGBM, XGBoost, TabFPN, and Decision Tree classifiers using demographic and temporal clinical variables, with or without HRV features. For ElasticNet, the regularisation parameters were tuned using Bayesian optimisation; for tree-based models, we used fixed or lightly tuned hyperparameters to reflect practical clinical deployment. Foundation model embeddings were extracted using PaPaGei for PPG signals and NormWear for ECG, each in zero-shot mode without downstream fine-tuning. PaPaGei embeddings were reduced using PCA and passed to a calibrated Random Forest classifier. NormWear outputs were averaged across segments per patient. All comparisons used expert-verified CVD status as ground truth.

## Model Implementation

To evaluate CVD detection from physiological time series under both zero-shot and label-efficient regimes, we employed two foundation models: NormWear[24] and PaPaGei[25]. The choice of two models reflects practical constraints: PaPaGei is the only publicly available foundation model specifically pretrained on PPG signals and represents the optimal choice for PPG-based screening. NormWear, pretrained primarily on ECG, serves two purposes: (1) providing native-modality ECG embeddings to evaluate whether ECG-derived features augment clinical data, and (2) establishing a zero-shot baseline for PPG (without any classifier training) to define the performance floor for truly label-free deployment, despite cross-modal limitations. We cannot use PaPaGei alone as it is PPG-specific and does not provide zero-shot classification; we cannot use NormWear alone as it is suboptimal for PPG. Using both models allows comprehensive evaluation across modalities and deployment scenarios.

NormWear was applied to raw three-lead ECG to extract embeddings for augmenting clinical features. For PPG, we evaluated a strictly zero-shot baseline using NormWear applied without any local training to produce patient-level probabilities, and a label-efficient pipeline using PaPaGei as a frozen encoder to produce embeddings that were fed to a lightweight calibrated Random Forest classifier. NormWear is included for two reasons: as a modality-matched comparator for ECG when augmenting clinic data without fine-tuning, and as a zero-fine-tuning PPG baseline that reflects the most label- and compute-constrained deployment scenario. This prevents cherry-picking and defines a lower bound that more practical, label-efficient pipelines must exceed. PCA on raw PPG served as an unsupervised feature baseline.

In contrast, PaPaGei was used to extract general-purpose embeddings from short PPG segments in a frozen-encoder setting. An overview of this pipeline is shown in Fig. 1. Raw PPG waveforms sampled at 100 Hz were segmented into overlapping 10 s windows (stride = 5 s). Each window underwent preprocessing, including denoising, detrending, z-score normalisation, and signal quality filtering. Only windows passing noise and flatline thresholds were retained. These were then passed to the PaPaGei foundation model via prompt-based inference to produce fixed-length physiological embeddings Table 1.

For each patient, embeddings across windows were mean-pooled and reduced via principal component analysis (PCA, retaining 15 components) to form a compact patient-level representation. These representations were input to a Random Forest classifier trained to predict CVD status. Classifier calibration was performed using isotonic regression on the training set. Evaluation was conducted on a stratified test set across 10 bootstrap iterations. Performance metrics, including AUROC, average precision, recall, precision, and F1 score, are reported in Table 2.

To interpret the learned representations, we projected the PCA-reduced embeddings using PCA, t-SNE, and UMAP, and visualised them with respect to ground truth labels and clinically important covariates. As shown in Figs. 3 and 4, PaPaGei embeddings exhibited strong class separability and aligned with key clinical predictors such as antiretroviral regimen and lipid profile changes. These results suggest that PaPaGei captures physiologically relevant structure from wearable PPG in a zero-shot setting.

## Table 1 | Baseline demographic and clinical features stratified by cardiologist-confirmed CVD status

| Feature | Overall (N=80) | CVD = 1 (N=13) | CVD = 0 (N=67) |
|---|---|---|---|
| Age (years) | 42 (35, 49) | 47 (44, 56) | 38 (28, 44) |
| Male sex, n (%) | 57 (71%) | 12 (92%) | 45 (67%) |
| Systolic BP (mmHg) | 120 (110, 130) | 130 (120, 140) | 110 (100, 120) |
| Hypertension treatment, n (%) | 3 (3.8%) | 2 (15%) | 1 (1.5%) |
| Current smoker, n (%) | 21 (26%) | 9 (69%) | 12 (18%) |
| Diabetes, n (%) | 3 (3.8%) | 3 (23%) | 0 (0%) |
| Family CVD history, n (%) | 50 (63%) | 10 (77%) | 40 (60%) |

Values are median (IQR) or n (%). Percentages are column-wise.

The preprocessing pipelines differed substantially between NormWear and PaPaGei applications, reflecting both each model's design specifications and our evaluation objectives. For ECG, NormWear was applied to three Shimmer-recorded leads (Lead I, Lead II, chest lead Vx) in its native modality with standard embedding extraction. For PPG, NormWear (zero-shot baseline) used minimal preprocessing, simple 128-sample windowing with 64-sample stride at the original sampling rate, no resampling, and no advanced filtering, to represent the most computationally constrained deployment scenario where neither local training nor intensive signal conditioning is feasible. In contrast, PaPaGei preprocessing was model-specific and more extensive: z-score normalisation, filtering and denoising, segmentation into 10 s windows, resampling from 64 Hz to 125 Hz, and padding to 1,250 samples, all designed to align with PaPaGei's documented input requirements for PPG. These preprocessing differences, combined with the cross-modal limitation of applying an ECG-pretrained model (NormWear) to PPG signals, mean that the NormWear PPG baseline

## Table 2 | Performance of clinical, supervised, and zero-shot models for CVD classification

| Model | Acc* | Recall | F1 Score | AUROC | AP Score |
|---|---|---|---|---|---|
| **Clinical only** | | | | | |
| TabFPN | 0.708 ± 0.10 | 0.500 ± 0.20 | 0.519 ± 0.19 | 0.667 ± 0.11* | 0.412 ± 0.15* |
| ElasticNet | 0.701 ± 0.09 | 0.500 ± 0.19 | 0.500 ± 0.17 | 0.705 ± 0.09** | 0.340 ± 0.13 |
| LightGBM | 0.701 ± 0.10 | 0.500 ± 0.21 | 0.500 ± 0.17 | 0.667 ± 0.10* | 0.362 ± 0.12 |
| XGBoost | 0.701 ± 0.09 | 0.500 ± 0.18 | 0.500 ± 0.16 | 0.641 ± 0.09* | 0.421 ± 0.14* |
| Random Forest | 0.779 ± 0.08 | **1.000 ± 0.00** | 0.469 ± 0.16 | 0.744 ± 0.08** | 0.433 ± 0.11** |
| Decision Tree | 0.543 ± 0.12 | 0.250 ± 0.20 | 0.317 ± 0.16 | 0.513 ± 0.13 | 0.210 ± 0.11 |
| Framingham† | 0.480 ± 0.10 | 0.500 ± 0.18 | 0.231 ± 0.15 | 0.551 ± 0.10 | 0.208 ± 0.12 |
| D:A:D Score† | 0.497 ± 0.11 | 0.500 ± 0.17 | 0.200 ± 0.13 | 0.462 ± 0.12 | 0.210 ± 0.11 |
| **Clinical + HRV features** | | | | | |
| TabFPN | 0.677 ± 0.10 | 0.500 ± 0.21 | 0.444 ± 0.17 | 0.715 ± 0.09* | 0.410 ± 0.14* |
| ElasticNet | 0.677 ± 0.11 | 0.500 ± 0.20 | 0.444 ± 0.18 | 0.622 ± 0.11 | 0.321 ± 0.12 |
| LightGBM | 0.500 ± 0.00 | 0.000 ± 0.00 | 0.000 ± 0.00 | 0.745 ± 0.10* | 0.444 ± 0.11* |
| XGBoost | 0.677 ± 0.10 | 0.500 ± 0.19 | 0.444 ± 0.17 | 0.692 ± 0.09* | 0.328 ± 0.13 |
| Random Forest | 0.631 ± 0.10 | 0.500 ± 0.18 | 0.364 ± 0.16 | 0.675 ± 0.10* | 0.379 ± 0.12 |
| Decision Tree | 0.543 ± 0.12 | 0.250 ± 0.17 | 0.317 ± 0.15 | 0.551 ± 0.13 | 0.208 ± 0.11 |
| **Clinical + ECG embeddings** | | | | | |
| TabFPN | 0.616 ± 0.10 | 0.500 ± 0.18 | 0.348 ± 0.16 | 0.667 ± 0.10* | 0.395 ± 0.13* |
| ElasticNet | 0.677 ± 0.09 | 0.500 ± 0.18 | 0.444 ± 0.17 | 0.705 ± 0.09** | 0.340 ± 0.13 |
| LightGBM | 0.500 ± 0.00 | 0.000 ± 0.00 | 0.000 ± 0.00 | 0.756 ± 0.10* | 0.322 ± 0.13 |
| XGBoost | 0.500 ± 0.00 | 0.000 ± 0.00 | 0.000 ± 0.00 | 0.667 ± 0.11* | 0.460 ± 0.13* |
| Random Forest | 0.555 ± 0.11 | 0.500 ± 0.17 | 0.286 ± 0.14 | 0.578 ± 0.12 | 0.291 ± 0.11 |
| Decision Tree | 0.497 ± 0.12 | 0.250 ± 0.17 | 0.240 ± 0.13 | 0.513 ± 0.12 | 0.192 ± 0.10 |
| **PPG-derived representations** | | | | | |
| PCA[u] | 0.528 ± 0.11 | 0.250 ± 0.12 | 0.300 ± 0.12 | 0.689 ± 0.10 | 0.208 ± 0.10 |
| NormWear (PPG)[z‡] | 0.552 ± 0.10 | 0.250 ± 0.12 | 0.333 ± 0.14 | 0.560 ± 0.09 | 0.226 ± 0.10 |
| NormWear (PPG)[d‡] | 0.677 ± 0.09 | 0.500 ± 0.18 | 0.333 ± 0.14 | 0.667 ± 0.08 | 0.322 ± 0.11 |
| **PaPaGei[c]** | **0.903 ± 0.06** | **1.000 ± 0.00** | **0.560 ± 0.16** | **0.769 ± 0.07**\*\* | **0.489 ± 0.12**\*\* |

Models were trained or evaluated using clinical features, ECG embeddings, or raw PPG signals. NormWear was used in zero-shot mode and PaPaGei embeddings were evaluated using a downstream classifier. Metrics are reported as mean ± SD over 10 bootstrap resamples. Framingham and D:A:D scores are included solely to characterise the study population's baseline cardiovascular risk profile; as these scores predict future events (5−10 year risk) rather than detect current abnormalities, they are not appropriate performance benchmarks for our diagnostic tool. Values in bold are the best for that metric. Acc* stands for Balanced Accuracy. u = unsupervised feature extraction followed by supervised classifier; z = zero-shot inference without supervised classifier; d = zero-shot inference with supervised classifier; c = pretrained embeddings followed by supervised classifier. Asterisks on PaPaGei indicate statistically significant improvement over Random Forest (Clinical only) at $p < 0.05$* (Wilcoxon signed-rank test) and $p < 0.01$**, though the modest effect size (AUROC difference 0.025) and small sample require cautious interpretation. † Framingham and D:A:D scores characterise baseline population risk for future events; not performance benchmarks for current CVD detection.
‡ NormWear PPG represents a multiply constrained baseline: cross-modal application (ECG model on PPG), minimal preprocessing (128-sample windows, no resampling), and zero local training. This establishes a performance floor for resource-limited deployment rather than optimal PPG model capabilities.

reflects multiple constraints: suboptimal modality matching, minimal signal conditioning, and zero local training. This combination establishes a conservative lower bound for resource-limited, zero-shot deployment rather than representing NormWear's optimal capabilities or a fair comparison with PPG-specific models.

## Evaluation

All models were trained and evaluated using stratified 80/20 train-test splits to preserve the proportion of cardiovascular disease (CVD) cases in both sets. To assess model stability and account for small sample variation, we generated 10 bootstrap resamples of the test set. We acknowledge that stratified k-fold cross-validation (e.g., 5-fold) would be methodologically preferable, providing non-overlapping test sets of equivalent size whilst ensuring each sample is evaluated exactly once. Bootstrap resampling with replacement provides valid variance estimates for the chosen evaluation strategy. Future larger validation studies will employ k-fold CV for cleaner evaluation partitioning. Model performance was reported as the mean and standard deviation of metrics across these resamples.

For models with tunable hyperparameters, including ElasticNet and tree-based methods, we conducted internal validation using stratified 3-fold cross-validation on the training split. ElasticNet hyperparameters (penalty and mixing ratio) were optimised via Bayesian optimisation to maximise the area under the receiver operating characteristic curve (AUROC). This approach adaptively selects hyperparameter combinations to explore based on surrogate modelling of performance, offering efficiency over grid or random search.

We evaluated model discrimination using the AUROC and average precision (AP), and binary classification performance using precision, recall, and F1 score. Let $\widehat{y}_i \in \{0, 1\}$ denote the predicted label and $y_i \in \{0, 1\}$ the ground truth for sample $i$. Then:

$$\text{Precision} = \frac{\text{TP}}{\text{TP} + \text{FP}} \qquad (1)$$

where TP is the number of true positives and FP is the number of false positives.

$$\text{Recall} = \frac{\text{TP}}{\text{TP} + \text{FN}} \qquad (2)$$

where FN is the number of false negatives.

$$\text{F1} = 2 \cdot \frac{\text{Precision} \cdot \text{Recall}}{\text{Precision} + \text{Recall}} \qquad (3)$$

AUROC (Area Under the Receiver Operating Characteristic Curve) quantifies the ability to rank positive instances above negative ones across all classification thresholds. It is equivalent to the probability that a randomly chosen positive instance has a higher predicted score than a randomly chosen negative one.

$$\text{AP} = \sum_n (R_n - R_{n-1}) \cdot P_n \qquad (4)$$

where $P_n$ and $R_n$ are the precision and recall at threshold $n$.

For statistical comparison of model performance, we conducted pairwise Wilcoxon signed-rank tests on AUROC and AP scores across the 10 test resamples. This non-parametric test accounts for paired, non-normally distributed data and is appropriate for small-sample comparative evaluation. Statistical significance was defined as $p < 0.05$, with results annotated accordingly in Table 2.

## Results

### Dataset

We conducted a prospective study at the Hospital for Tropical Diseases in Ho Chi Minh City, Vietnam, enroling 80 adults living with HIV (PLWH) between November 2023 and July 2024. All participants received routine antiretroviral therapy and had no acute illness at enrolment. Each participant was assessed for cardiovascular risk using Framingham and D:A:D-modified Framingham scores, and for evidence of cardiovascular disease using a standard 12-lead electrocardiogram (ECG) and echocardiography reviewed by expert cardiologists. In addition, wearable photoplethysmography (PPG) monitoring was performed using the SmartCare device. PPG waveforms were recorded at 100 Hz for ~20 min and then segmented into 5 min windows. Conventional HRV and waveform morphology features were extracted for supervised baselines; our proposed approach uses short raw PPG segments with minimal preprocessing and no encoder fine-tuning. Follow-up visits occurred approximately every 3 months, with repeated PPG and ECG measurements.

We used binary labels indicating the presence or absence of cardiologist-confirmed CVD based on ECG and echocardiography interpretations. Participants were labelled CVD = 1 if clinically significant abnormalities were observed in either modality, and CVD = 0 otherwise. This binary label serves as ground truth for model evaluation; Framingham and D:A:D risk scores were retained as continuous comparators (see Methods). The 13 CVD cases exhibited a range of cardiovascular abnormalities identified through ECG and echocardiographic assessment. Abnormalities included cardiac structural changes (left ventricular hypertrophy, left atrial enlargement), ECG patterns suggestive of myocardial ischaemia, conduction system abnormalities (bundle branch blocks, AV block), and arrhythmias (ventricular premature contractions). Several participants exhibited multiple coexisting abnormalities, representing more advanced cardiovascular involvement. These findings ranged from isolated structural changes to complex presentations requiring further cardiological evaluation, illustrating the spectrum of CVD severity that our screening approach aims to detect in asymptomatic PLWH. We note that this heterogeneous mix of abnormalities reflects a broad CVD detection target rather than specific prediction of coronary artery disease, and that subclinical coronary atherosclerosis without ECG or echocardiographic manifestations would not be captured by our ground truth definition. These features were used as inputs for supervised baselines, with ECG or echocardiographic evidence of CVD over 1 year serving as ground truth.

ECG has < 50% sensitivity for detecting prior myocardial infarction[29] and echocardiography ~61%[30], meaning subclinical coronary artery disease (CAD) without prior MI would be missed by our approach. Furthermore, our definition captures a heterogeneous spectrum of cardiovascular abnormalities–including structural changes (left ventricular hypertrophy, atrial enlargement), electrical abnormalities (conduction disease, arrhythmias), and patterns suggestive of ischaemia–that extends beyond the atherosclerotic CVD that Framingham and D:A:D scores primarily predict. This creates a mismatch between our detection target (prevalent abnormalities of any cardiovascular origin) and traditional risk prediction endpoints (incident atherosclerotic events, particularly myocardial infarction and stroke). In the absence of coronary angiography or cardiac CT (not feasible in this resource-constrained outpatient setting), we used the highest-fidelity cardiovascular assessments available, cardiologist-interpreted 12-lead ECG and transthoracic echocardiography, while acknowledging these ground truth limitations.

### Baseline cardiovascular risk profile

To characterise the baseline cardiovascular risk in this cohort, we calculated Framingham Risk Scores for general cardiovascular disease and D:A:D-modified Framingham scores for all participants. Using standard risk categories, 52 participants (65%) were classified as low risk ( < 10% 10 year risk) and 28 (35%) as intermediate-to-high risk (≥10% 10 year risk) according to the Framingham score. For the D:A:D-modified Framingham score, 36 participants (45%) were classified as low risk and 44 (55%) as

moderate-to-very-high risk. These prognostic scores predict future cardiovascular events over 5−10 years and are reported here to characterise the study population, not to benchmark diagnostic performance for detecting current CVD abnormalities.

## Baseline CVD classification using clinical features

We first evaluated the ability of supervised classifiers to detect ECG or echocardiographic evidence of CVD over 1 year using only static and summarised clinical variables collected over four outpatient visits, including age, sex, blood pressure, and cholesterol. Models were evaluated on a stratified held-out test set using 10 bootstrap resamples to estimate variability. As summarised in Table 2, Random Forest achieved the highest AUROC (0.744 ± 0.08) and average precision (0.433 ± 0.11) among supervised models using clinical features. However, all of the variables included in the Random Forest are also included in the traditional risk scores. Traditional risk scores showed limited discrimination for prevalent CVD in this cohort (Framingham AUROC 0.551; D:A:D 0.462). This is expected: the scores are designed to estimate future event risk, not to detect contemporaneous abnormalities. Here, we use them as continuous comparators for triage-style ranking; even under this generous interpretation, they underperformed the embedding-based pipeline.

TabFPN yielded the highest precision (0.538 ± 0.21) and F1 score (0.519 ± 0.19), indicating strong discriminative ability. Random Forest, however, achieved perfect recall (1.000 ± 0.00), maximising sensitivity to CVD cases at the cost of lower precision (0.306 ± 0.17). In contrast, traditional risk scores performed poorly, with AUROCs of 0.551 (Framingham) and 0.462 (D:A:D), highlighting their limited applicability in this HIV-positive outpatient population. ElasticNet offered a good trade-off between discrimination and interpretability, with balanced metrics and an AUROC of 0.705 ± 0.09. Overall, these results show that even in low-resource settings, simple supervised models trained on routine outpatient data can substantially improve CVD stratification, particularly compared to imported risk scoring systems.

## Foundation model augmentation with ECG embeddings

This analysis uses NormWear in its intended modality (ECG), extracting embeddings from three Shimmer-recorded leads (Lead I, Lead II, chest lead Vx), avoiding the cross-modal and preprocessing constraints present in the zero-shot PPG baseline. Prompt templates for all foundation models used are detailed in the Supplementary Materials section 3.

We next examined whether adding ECG-derived foundation-model representations to routine clinic data improves CVD prediction under label-limited conditions. Patient-level embeddings were extracted from raw 12-lead ECG using a frozen encoder (NormWear), reduced with PCA, and concatenated to static and temporal clinical features; no fine-tuning of the encoder was performed. NormWear primarily targets ECG but can produce zero-shot scores from PPG; we use it as a baseline without local training. This design tests whether a costlier signal (ECG) and a strong pretrained encoder improve deployable screening without fine-tuning in a small, imbalanced cohort of PLWH. PaPaGei is specialised for PPG and was not applied to ECG.

As summarised in Table 2, effects were model-dependent. ElasticNet remained stable relative to clinical-only inputs (Balanced Accuracy 0.677 ± 0.09, F1 0.444 ± 0.17). In contrast, tree ensembles frequently exhibited a small-$n$ failure mode: LightGBM attained a high AUROC (0.756 ± 0.10) but defaulted to no positive calls (recall 0.000 ± 0.00; Balanced Accuracy 0.500 ± 0.00), with a similar pattern for XGBoost. The divergence between ranking (AUROC) and operating-point metrics (recall, Balanced Accuracy) indicates that while ECG embeddings contain a discriminative signal, a label-efficient alignment step is required to convert ranking into calibrated decisions at this scale.

These findings suggest limited incremental benefit from zero-fine-tuning ECG encoders over clinic data in our cohort, whereas the strongest gains arise in the PPG pathway when paired with a lightweight local calibrator.

## CVD detection using pretrained embeddings with lightweight calibration

We evaluated whether pretrained physiological representations from wearable PPG can support cardiovascular disease detection (without any additional clinical data) under two deployment modes: a strictly zero-shot baseline, and a label-efficient pipeline with a frozen encoder and a lightweight local classifier. In the zero-shot setting, NormWear was applied without any local training to generate patient-level CVD probabilities from short PPG windows. We explicitly acknowledge three methodological constraints in this baseline: (1) NormWear was primarily designed and pretrained for ECG, and its application to PPG represents cross-modal transfer that may not fully capture PPG-specific physiological characteristics such as pulse wave morphology, (2) minimal preprocessing was applied–simple 128-sample windowing (stride 64) at the original sampling rate without resampling, z-score normalisation, or advanced filtering–representing a computationally constrained scenario rather than optimal signal conditioning, and (3) no local training or calibration was performed, establishing a strictly zero-shot, label-free deployment. These constraints were intentional to establish a performance floor representing the most resource-limited deployment scenario where neither task-specific models, intensive preprocessing, nor local labels are available. As shown in Table 2, zero-shot performance was modest (AUROC 0.560, AP 0.226; Balanced Accuracy 0.552, F1 0.333), indicating that while some physiologically relevant signal is captured, discrimination is limited under these multiply constrained conditions.

In the label-efficient setting, PaPaGei was used as a frozen encoder to produce PPG embeddings, which were then passed to a calibrated Random Forest classifier trained on the local cohort. Given the class imbalance (13 CVD cases among 80 participants, 16% prevalence), we report both AUROC and AUPRC (average precision, equivalent to area under the precision-recall curve). AUPRC is generally more informative than AUROC for imbalanced classification tasks, as it directly reflects the precision-recall trade-off without being dominated by the large negative class. PaPaGei's AUPRC of 0.489 ± 0.12 substantially exceeds the baseline expectation of 0.163 (the prevalence rate) and outperforms all comparator methods, indicating meaningful discriminative ability beyond simply predicting the majority class. At the selected operating threshold (calibrated via isotonic regression), PaPaGei achieved sensitivity 1.00 (all CVD cases detected), specificity 0.70 (47 of 67 CVD-negative cases correctly classified), positive predictive value 0.39 (13 of 33 positive predictions correct), and negative predictive value 1.00 (no CVD cases among negative predictions). The complete confusion matrix metrics are provided in Supplementary Table 2. However, these performance estimates have substantial uncertainty due to the small positive class size (13 CVD cases across 10 bootstrap resamples), and bootstrap resampling, while providing variance estimates, does not overcome fundamental small-sample limitations or guarantee stable performance in independent cohorts. We acknowledge that stratified k-fold cross-validation (e.g., 5-fold) would be methodologically preferable, providing non-overlapping test sets of equivalent size and ensuring each sample is evaluated exactly once. Bootstrap resampling with replacement may introduce sampling bias, though it provides valid variance estimates for the chosen evaluation strategy. Future validation studies should employ k-fold CV for cleaner evaluation partitioning. PaPaGei achieved perfect recall (sensitivity), meaning no CVD cases were missed in our pilot evaluation, a key requirement for a prescreening tool where the clinical cost of false negatives (missed CVD cases) substantially exceeds that of false positives (unnecessary confirmatory testing). The moderate F1 score reflects this trade-off: while precision was lower (~0.39, derived from F1 and recall), the false positives would simply undergo confirmatory ECG and echocardiography, a clinically acceptable pathway for ensuring comprehensive case detection in this high-risk population. Notably, this improvement was achieved without any fine-tuning of the foundation model itself.

However, these performance estimates have substantial uncertainty due to the small positive class size (13 CVD cases across 10 bootstrap resamples), and bootstrap resampling, while providing variance estimates,

**Fig. 2 | Predicted CVD likelihood distributions across models stratified by CVD label.** We visualised predicted CVD probabilities for each model using violin plots split by ground-truth status (No CVD = blue, CVD = red). Traditional clinical risk scores (Framingham, D:A:D-modified Framingham) showed limited separation with overlapping distributions and compressed ranges. Note: Framingham and D:A:D scores are shown for reference as population descriptors; these prognostic scores predict future cardiovascular events and are not appropriate benchmarks for our diagnostic tool detecting current abnormalities. A supervised Random Forest trained on clinical features improved separation, particularly for low-risk individuals, but substantial overlap remained. In contrast, the PaPaGei + calibrator pipeline (frozen pretrained encoder with a lightweight local classifier) demonstrated a pronounced rightward shift for CVD-positive individuals, concentrating high predicted risk among true cases and indicating enhanced discriminative power from wearable PPG without fine-tuning the foundation model.

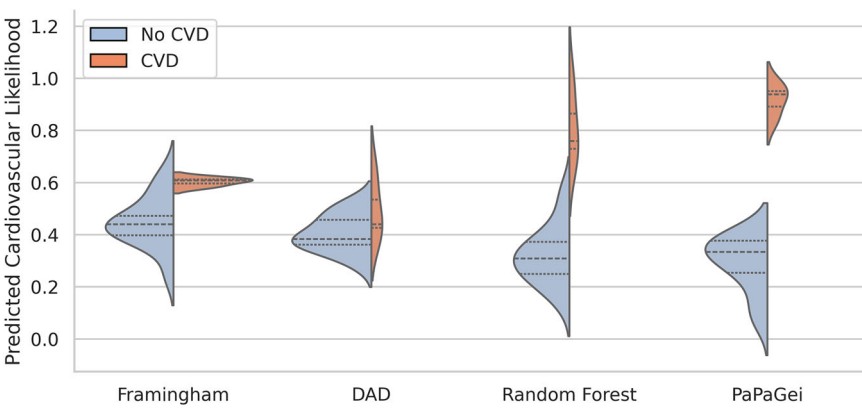

does not overcome fundamental small-sample limitations or guarantee stable performance in independent cohorts. These results should be interpreted as proof-of-concept evidence warranting further investigation rather than definitive validation of deployment readiness.

As an unsupervised feature baseline, we applied principal component analysis (PCA) directly to raw PPG segments to obtain patient-level representations, followed by a Random Forest classifier. PCA retained some ranking ability (AUROC 0.689 ± 0.10), but class balance remained poor (Balanced Accuracy 0.528 ± 0.11, recall 0.250 ± 0.12; F1 0.300 ± 0.12), consistent with the low-signal, imbalanced setting and the absence of task-specific representation learning. Together, these results underscore the added value of pretrained PPG embeddings, with a minimal, on-site calibrator materially improving discrimination over both zero-shot scoring and unsupervised features.

To characterise how models stratify CVD, we visualised predicted probability distributions by ground-truth label (Fig. 2). Framingham and D:A:D showed narrow, overlapping distributions, while a supervised Random Forest trained on clinical features improved separation primarily for low-risk patients. The PaPaGei + calibrator pipeline exhibited the clearest separation, concentrating high predicted likelihood of CVD almost exclusively among true CVD-positive cases, reflecting better calibration and sensitivity from raw PPG inputs.

### Embedding separability and feature attribution

To better understand the structure of the learned patient representations, we visualised the embeddings produced by each model input type using three standard dimensionality reduction methods: PCA, UMAP, and t-SNE. As shown in Fig. 3, patient-level features derived from clinical variables showed minimal class separability between individuals with and without cardiovascular disease (CVD). NormWear embeddings extracted from raw ECG signals provided marginal improvement, though the overlap between classes remained substantial. In contrast, PaPaGei embeddings computed directly from short PPG waveform segments demonstrated markedly clearer class separation across all projection methods. These findings suggest that PaPaGei captures richer, task-relevant physiological structure from raw PPG in a zero-shot setting, despite the absence of training on cardiovascular outcomes or ECG labels. PaPaGei has reported competitive performance on related PPG tasks, including cuffless blood pressure estimation and hypertension screening, suggesting its embeddings could capture haemodynamic structure relevant to our CVD-detection application[25]. This supports the potential utility of foundation model embeddings as standalone

features for CVD probability stratification, particularly when traditional clinical features or labelled training data are limited.

To further investigate whether PaPaGei embeddings encode clinically meaningful signals, we overlaid selected clinical variables, previously identified as important by supervised models, onto the projected embedding space (Fig. 4). Notably, patients receiving the antiretroviral regimen Tenofovir/lamivudine/dolutegravir (TDF_3TC_DTG) appeared largely confined to low-risk regions of the latent space, supporting its observed protective association with CVD absence. Conversely, larger changes in total cholesterol across visits (TOTAL_CHOL_delta), a known cardiovascular risk marker, were concentrated in high-risk clusters where CVD-positive patients were more prevalent. These overlays reveal alignment between PaPaGei's unsupervised physiological representations and clinically validated risk factors, reinforcing the model's capacity to capture latent structure that is both discriminative and biologically informative. This interpretability also demonstrates how foundation models may serve as hypothesis-generating tools, highlighting patterns of treatment response or physiological dysregulation that merit further investigation.

Training details for all models can be found in Section 2 of the Supplementary Material.

## Discussion

In this proof-of-concept pilot study (N = 80 PLWH, 13 CVD cases), we evaluated whether pretrained physiological embeddings from foundation models can support cardiovascular screening when combined with lightweight local calibration. We compared two pragmatic scenarios: strictly zero-shot deployment (NormWear) and a hybrid approach using frozen PaPaGei embeddings with a locally-trained classifier. In this pilot cohort, the PaPaGei pipeline achieved encouraging performance (AUROC 0.769), numerically higher than the zero-shot baseline, unsupervised PCA features, and supervised models using clinical features alone. Our PPG-based diagnostic approach and traditional prognostic risk scores (Framingham, D:A:D) address fundamentally different clinical questions and are therefore not directly comparable. Risk scores predict future cardiovascular events over 5−10 years to guide preventive interventions, whereas our method aims to detect current cardiovascular abnormalities for triage to confirmatory testing. We therefore focus comparisons on other diagnostic approaches: zero-shot foundation models, unsupervised feature extraction (PCA), and supervised models trained on the same clinical variables. However, these preliminary findings require validation in larger, multi-centre cohorts before clinical deployment.

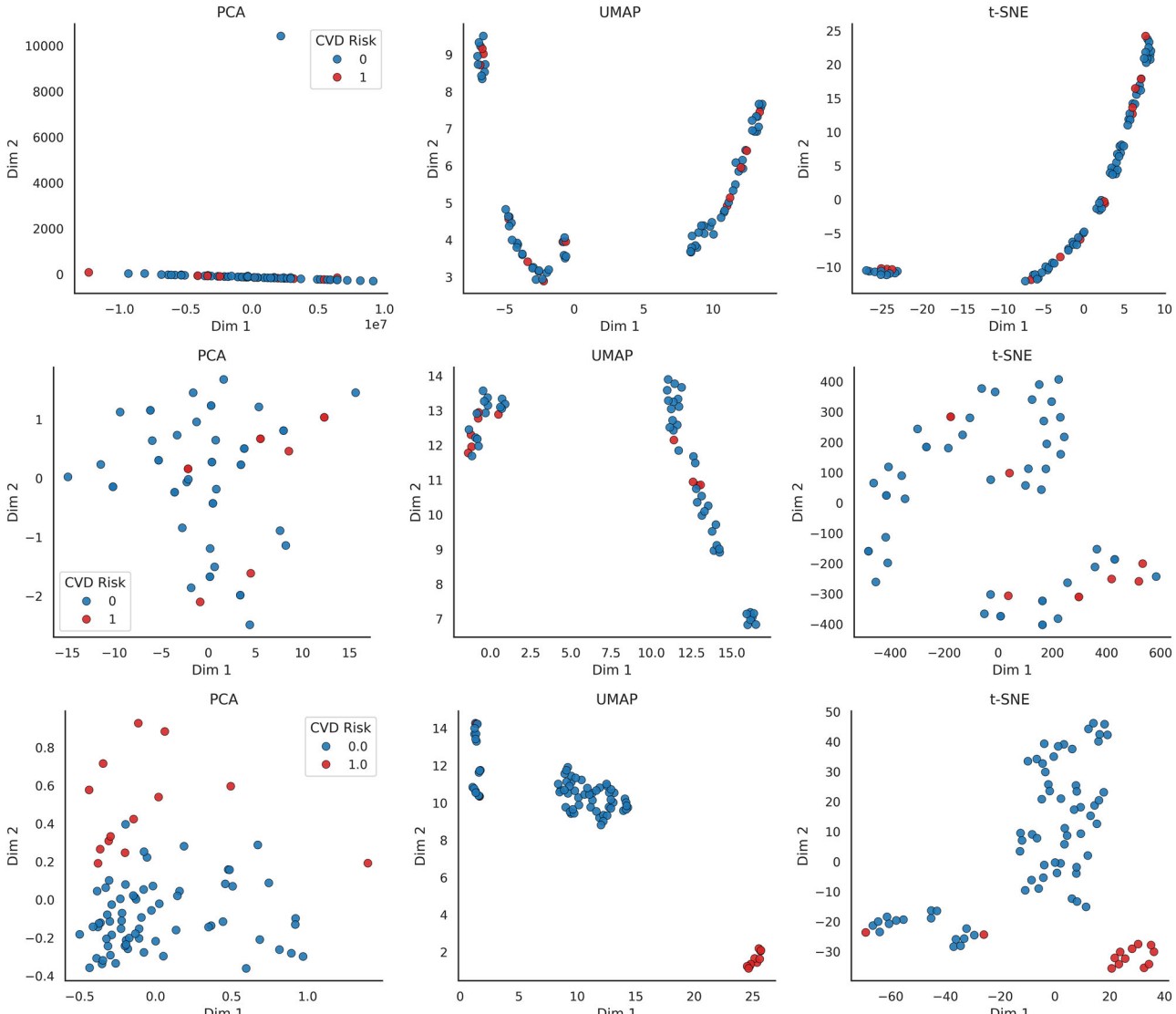

**Fig. 3 | Latent space projections of patient representations by model input type.** We visualised the separability of patient-level CVD likelihood using PCA, UMAP, and *t*-SNE projections on (**a**) routine clinical features, (**b**) NormWear embeddings, and (**c**) PaPaGei PPG embeddings. Compared to clinical and ECG-derived representations, PaPaGei embeddings showed markedly better class separation between CVD and non-CVD patients across all dimensionality reduction methods.

Compared to clinical and ECG-derived representations, PaPaGei embeddings showed markedly better class separation between CVD and non-CVD participants across all dimensionality reduction methods. This suggests that PaPaGei captures richer physiological signals relevant to cardiovascular status from raw PPG with a frozen encoder (no fine-tuning).

Importantly, while our approach required local labels to train the classifier, it avoided the computational burden of fine-tuning the foundation model itself. The PaPaGei embeddings remained frozen throughout, preserving their pretrained representations while a simple Random Forest learned to map these features to cardiovascular disease likelihood. The clinical plausibility of the learnt latent structure further supports deployment. PaPaGei embeddings organised patients on dolutegravir-based regimens into low-risk regions, aligning with evidence for cardiometabolic benefits of newer antiretrovirals[31–33]. Conversely, greater visit-to-visit cholesterol variability concentrated in high-risk regions, consistent with known associations between lipid instability and cardiovascular events in HIV[34,35]. These patterns emerged without any HIV-specific fine-tuning, indicating that pretrained representations capture physiology that transfers across populations and care contexts.

From an implementation perspective, this embedding-based approach shows potential as a practical middle ground. While not truly "zero-shot," it may reduce the complexity compared to training deep models from scratch. However, the generalisability of this approach beyond our pilot cohort remains to be established. The foundation model handles feature extraction, requiring only a lightweight classifier to be trained locally. This could be accomplished with standard computing resources available in most clinical settings, unlike the GPU infrastructure needed for deep learning. We emphasise that PPG is not a clinician-interpreted signal; the utility here derives from embeddings that encode physiological structure and support triage decisions when paired with a shallow calibrator. Our best approach still requires labelled data to train the local classifier. Our contribution addresses *computational* constraints (avoiding GPU infrastructure and encoder fine-tuning) rather than label scarcity. This suits contexts where labels can be obtained through existing clinical workflows but computational resources are limited. However, we have not characterised label efficiency, minimum sample sizes and performance scaling with fewer labels remain unknown. Our zero-shot baseline (NormWear, AUROC 0.560) represents truly label-free deployment but yields insufficient discrimination, highlighting that some local labels substantially improve utility. Future work should include learning curve analyses and exploration of few-shot learning methods.

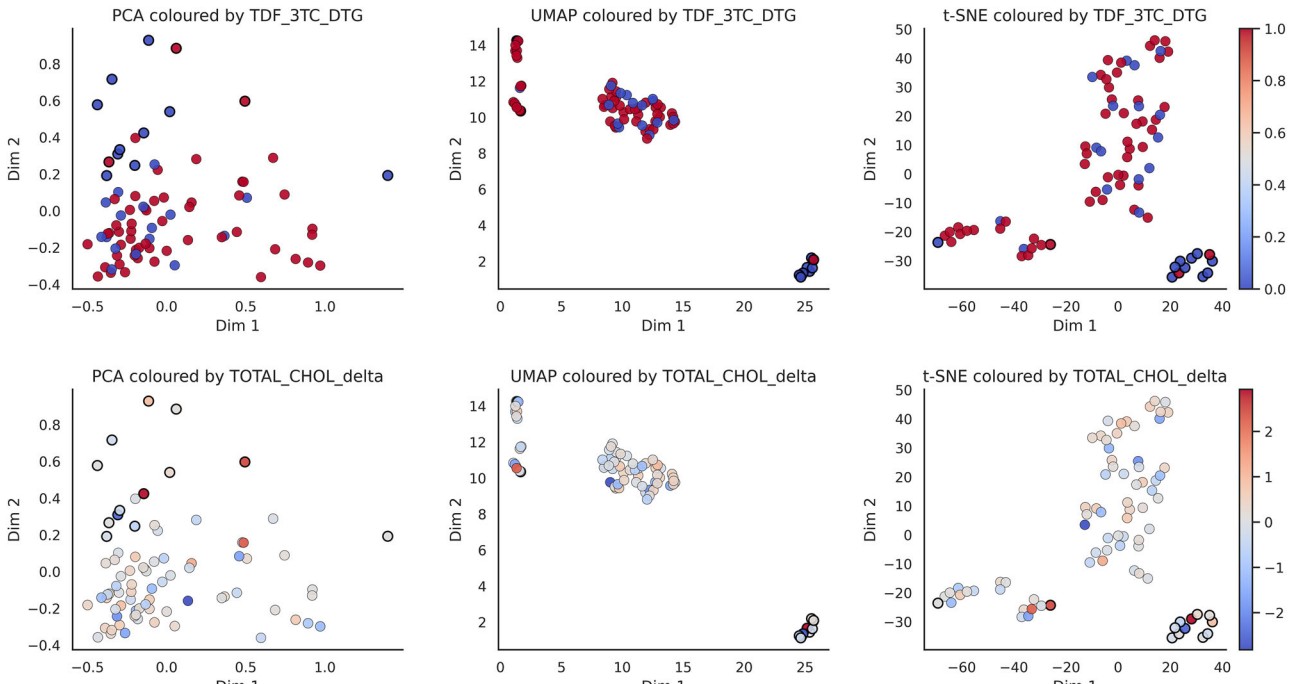

**Fig. 4 | Clinical variable overlays on PaPaGei PPG embedding space.** We visualised the projections of patient-level PaPaGei embeddings using PCA, UMAP, and t-SNE, overlaid with clinically relevant variables. Dot outlines indicate binary cardiovascular disease (CVD) status (black = CVD, white = no CVD), while colour represents values of selected clinical predictors. (**Top row**) TDF_3TC_DTG usage is associated with a lower density of CVD-positive patients, consistent with protective antiretroviral therapy effects. (**Bottom row**) Changes in total cholesterol (TOTAL_CHOL_delta) across visits are higher in the CVD-positive clusters, supporting its predictive relevance. These overlays confirm that clinical variables identified as important by supervised models align with separable physiological patterns learned by PaPaGei without encoder fine-tuning.

Our single-site cohort is extremely small ($N = 80$ with only 13 CVD cases), severely limiting statistical power, generalisability, and the reliability of performance estimates. This sample size is insufficient to establish definitive conclusions about model performance or to support clinical implementation. The class imbalance (16% CVD prevalence) compounds these issues, and while bootstrap resampling provides variance estimates, it cannot overcome the fundamental constraints of small sample size or create information not present in the original data. Performance metrics, particularly the observed perfect recall, likely represent optimistic estimates that may not generalise beyond this specific cohort. While bootstrap resampling provides uncertainty estimates, larger multi-site evaluations are needed to characterise stability across devices, acquisition conditions, and case mix[36]. Although PaPaGei was never fine-tuned on local outcomes, the use of a shallow calibrator means performance reflects a label-efficient rather than strictly label-free configuration. Finally, potential spectrum and selection biases, as well as choices in signal preprocessing, windowing, and embedding aggregation, may influence generalisability. Our comparison between current disease detection and likelihood (foundation models) and future risk prediction (clinical scores) represents different clinical endpoints.

The zero-shot NormWear PPG baseline embodies multiple methodological constraints that must be considered when interpreting performance: cross-modal application of an ECG-pretrained model to PPG signals, minimal preprocessing (128-sample windowing without resampling or advanced filtering), and zero local training. While these constraints were intentional, establishing a performance floor for the most resource-limited deployment where PPG-specific models, intensive signal processing, and local labels are all unavailable, they mean this baseline does not represent optimal PPG foundation model capabilities. In contrast, PaPaGei benefits from PPG-specific pretraining, model-appropriate preprocessing (z-score normalisation, filtering, resampling to 125 Hz, quality control), and lightweight local calibration (frozen encoder with trained classifier). The substantial performance difference (NormWear PPG AUROC 0.560 vs

PaPaGei AUROC 0.769) reflects both the value of modality-matched pretraining and the benefit of appropriate preprocessing and minimal local adaptation. The appropriate modality-matched comparisons are therefore NormWear on ECG (AUROC 0.705-0.756 when augmenting clinical features) and PaPaGei on PPG, with the NormWear-PPG results interpreted as a methodological lower bound rather than a fair head-to-head comparison.

The small sample size ($N = 80$, 13 CVD cases) represents the study's most critical limitation and fundamentally constrains the statistical foundation of our findings. With only 13 positive cases, our test set contained ~2–3 CVD cases per bootstrap iteration, resulting in substantial uncertainty in all performance estimates despite the bootstrap resampling strategy. While we report mean AUROC of 0.769 ± 0.07 (95% CI: 0.70, 0.84), the wide confidence interval reflects genuine uncertainty about model performance, not merely sampling variability that can be overcome through resampling. Small imbalanced datasets are prone to optimistic bias in performance estimation, even with careful cross-validation and bootstrap procedures. The reported metrics, particularly the perfect recall, may not generalise to larger, more heterogeneous populations. Furthermore, with limited positive cases, our ability to detect statistically significant performance differences between models is constrained, and comparisons showing $p < 0.05$ should be interpreted cautiously given the limited statistical power. We emphasise that this study provides hypothesis-generating, proof-of-concept evidence rather than confirmatory validation. The findings suggest that frozen foundation model embeddings warrant further investigation in adequately powered studies. For stable performance estimates and sufficient power to detect clinically meaningful differences, future validation studies should target larger sample sizes, ideally across multiple sites with diverse patient populations and care settings.

Our CVD ground truth definition based on ECG and echocardiography has important limitations in both sensitivity and specificity that warrant careful interpretation. The gold standard for CAD detection, invasive coronary angiography or coronary CT angiography, was not

clinically indicated or feasible in our asymptomatic outpatient cohort in this resource-constrained setting. Future validation studies should incorporate more definitive CAD endpoints where feasible, stratify model performance by CVD subtype (structural vs ischaemic vs electrical), and clarify whether PPG-derived embeddings generalise across the heterogeneous CVD spectrum or require subtype-specific calibration.

While this study was not a formal economic evaluation, we outline pragmatic considerations for deploying a wearable PPG + foundation-model pipeline as a prescreening tool in HIV outpatient care. Echocardiography (and clinician-interpreted ECG) provides structural and electrophysiological assessment that PPG cannot. Short PPG captures can be obtained in the waiting area by non-specialist staff or remotely via wearables and scored on commodity hardware (frozen encoder, shallow classifier), reducing dependence on laboratory testing and specialist time within a single outpatient encounter. A formal micro-costing study is an important next step to quantify these trade-offs.

Future work should prioritise prospective, multi-centre validation in Vietnam, calibration and threshold selection for operational triage, and cost-effectiveness modelling to balance screening coverage against referral capacity. Such economic evaluations should account for the differential costs of false-positive (confirmatory testing) versus false-negative (missed CVD with potential adverse outcomes) classifications, with sensitivity analyses exploring various operating thresholds and their impact on healthcare resource utilisation, patient outcomes, and overall system efficiency. Methodologically, we find two directions to be most promising: learning-curve analyses to quantify label efficiency (how performance scales from a handful of labels), and strictly label-free anomaly scoring over foundation-model embeddings (e.g., mixture modelling or distance-to-normative embeddings) as a complement to the lightweight calibrator. Fairness analyses across sex, age, and antiretroviral regimens are also important for real-world deployment.

In conclusion, this pilot study provides preliminary, hypothesis-generating evidence that pretrained physiological models may enable low-cost cardiovascular screening without task-specific fine-tuning and with only minimal local supervision, showing numerically higher discrimination compared with zero-shot baselines, unsupervised features, and supervised models using clinical data alone in this small proof-of-concept cohort. However, given the severe statistical limitations imposed by our sample size ($N = 80$, 13 CVD cases), these performance estimates should be interpreted with caution. Adequately powered validation in larger, prospective, multi-centre studies with diverse populations is essential before any clinical implementation can be considered[10,37]. Pending rigorous external validation, this work outlines a potential methodological framework for deploying AI-driven screening where labelled data and specialist resources are limited, which could support more equitable access to cardiovascular care for people living with HIV.

## Data availability

The datasets analysed in this study include sensitive patient-level clinical data, electrocardiography recordings, echocardiography assessments, and photoplethysmography signals from people living with HIV collected at the Hospital for Tropical Diseases, Ho Chi Minh City, Vietnam. These data cannot be made publicly available due to ethical and privacy restrictions. The data are subject to institutional policies of the Hospital for Tropical Diseases and Vietnamese governmental regulations regarding patient privacy and healthcare data. Individual-level data cannot be shared without additional ethical approval and patient consent, which were not obtained as part of the original study protocol. Researchers interested in collaboration or data access should contact the corresponding author, but access cannot be guaranteed due to the aforementioned restrictions. The source data for Figs. 2, 3, and 4 may be found in the appropriately labelled Supplementary Data files.

## Code availability

Code for running and analysing the models can be found here[38].

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

## Acknowledgements

M.M. is supported by the Rhodes Trust and the EPSRC CDT Health Data Science. T.Z. is supported by the Royal Academy of Engineering. L.T. is funded by the Wellcome Trust.

## Author contributions

M.M., H.H.B., L.T., and T.Z. conceived the study. M.M. developed the machine learning methodology, implemented the computational pipeline, performed statistical analyses, created visualisations, and drafted the manuscript. H.H.B. coordinated data collection, managed the clinical database, performed data quality control, and contributed to study design. L.V.T., V.N.Q., and X.H.V. provided clinical expertise, performed cardiovascular assessments, and interpreted cardiological findings. N.N.T., T.A.N.H., M.T.V.H., P.N.Q.K., P.V.H., K.L.D.V., and Y.L.M. recruited participants, conducted clinical assessments, collected physiological recordings, and managed patient follow-up. L.T. and T.Z. supervised the research, provided critical feedback, and secured funding. All authors reviewed and approved the final manuscript.

## Competing interests

We declare no competing interests.
