## [Transparent Peer Review file · Communications Medicine]

Foundation models enable wearable signal screening for cardiovascular disease among people living with HIV

Corresponding Author: Mr Munib Mesinovic

Version 0:

Reviewer comments:

Reviewer #1

(Remarks to the Author)

Overview: In this study, Mesinovic and colleagues evaluated pretrained physiological embeddings from foundation models for CVD detection in people with HIV (PWH) in Vietnam using wearable signals in the outpatient setting. Tools for detection and prevention of CVD are high-ranked among the current research priorities in the era of implementation science, particularly in PWH as their risk of CVD is nearly double of that in the general population. While combination antiretroviral treatment has increased the life expectancy of PWH to nearly that of people without HIV, the mortality gap due to CVD remains. Compared to CVD abnormalities per standard 12-lead electrocardiogram and echocardiography reviewed by expert cardiologists, the Authors' proposed wearable photoplethysmography is promising as a prescreening tool for triaging in resource-limited settings in order to identify those who may warrant confirmatory testing. The Authors were thorough evaluating a variety of models using training and test set and variety of performance metrics, and the manuscript is well-written. I have the following comments:

Major:

1. The analysis population consists of 80 PWH and only 13 of those were identified to have CVD abnormalities by expert cardiologists. With these limited numbers, the study is a proof-of-concept or pilot study. While training-test analyses were conducted, the prediction statistics with such small numbers, further split to preserve the proportion with CVD in the train-test set, are likely to be optimistic. The Authors appropriately mention this caveat and include multi-center validation as a priority for future work in the Discussion. However, the strong language at the beginning of the Discussion ("demonstrates", "superior performance", "substantially") is inconsistent with these limitations and misleading. Please consider softening the language to reflect the level of evidence and mention the numbers of cases (13) in the abstract.
2. The value of comparisons to the Framingham and D:A:D risk scores in this setting is questionable. As the Authors note, these were developed to assess the risk of future major cardiac events (over 10 or 5 years for Framingham and D:A:D, respectively) and are used to target preventive interventions whereas the proposed tool is evaluated for diagnostic use to identify participants for further confirmatory evaluation of current CVD abnormalities. In addition, the risk scores predict major CVD events such as CV death, MI, stroke and angina, although it is not clear which Framingham score was used (CHD from 1998, 2001 or the total CVD score), and the endpoints in each are somewhat different. As such, these scores do not seem appropriate comparisons for the proposed PaPaGei-embedding approach. I feel the manuscript would be clearer without comparisons to these risk scores without losing any value. The average Framingham and D:A:D risk scores could still be reported to characterize the study population at enrollment.
3. Building on the previous comment, the types of CVD abnormalities in the 13 cases would provide important context on the events that can be detected. Did they include a variety of different abnormalities, were they milder or major CVD events? Please consider including a brief summary of these 13 CVD abnormalities in the Results.
4. A variety of performance metrics are included. Could the Authors elaborate on which should be prioritized in this setting? For example, the proposed PaPaGei approach had estimated recall of 1 which means that no cases went undetected. While estimated precision was lower 0.56, a lower precision may be acceptable in this setting, since the cost of false positive is not as high as the cost of false negative. The former would lead to unnecessary confirmatory evaluation and use of resources when there is no CVD abnormalities whereas the latter to missing a true case which could have serious consequences to person's health and life. Elaborating on these considerations would help put the results in context, and provide direction for future economic cost-benefit evaluations thoughtfully mentioned by the Authors.

Minor:

5. In the supplement Table 1, please consider including a breakdown by cardiologist-confirmed CVD status, as in Table 1 in the manuscript.

Reviewer #2

(Remarks to the Author)

This paper investigates the use of pretrained physiological foundation models (PaPaGei and NormWear) for cardiovascular disease (CVD) screening in people living with HIV (PLWH) using wearable photoplethysmography (PPG) signals. The authors compare zero-shot and label-efficient approaches, finding that frozen PaPaGei embeddings with local classifier calibration achieved superior discrimination compared to traditional clinical risk scores and zero-shot baselines.

The work addresses an important clinical need, as cardiovascular risk assessment in PLWH populations is challenging and traditional risk scores systematically underperform. The emphasis on practical deployment in resource-limited settings is commendable, and the visualisation of treatment regimens and cholesterol variability in embedding space provides evidence for biological plausibility.

However, several methodological limitations undermine the strength of the conclusions:

The study's most critical limitation is the small sample size (N=80, 13 CVD cases). This severely limits generalisability beyond this specific cohort, statistical power for detecting true performance differences, and stability of performance estimates despite bootstrap resampling. The reported AUROC of 0.769, while seemingly promising, lacks sufficient statistical foundation given the small positive class size.

The heavy reliance on AUROC is problematic for such an imbalanced dataset. AUROC can be misleadingly optimistic when positive cases are rare. The authors should report area under precision-recall curve AUPRC, positive predictive value (PPV), and negative predictive value (NPV) to provide a more realistic assessment of clinical utility. The perfect recall (1.000 ± 0.00) reported for some models suggests potential overfitting to the small positive class - the models may be defaulting to positive predictions due to the extreme imbalance.

The ECG analysis suffers from some methodological issues. The cross-modal application of NormWear (designed for ECG) to PPG creates unfair comparisons, while insufficient detail is provided on ECG feature extraction and processing.

Comparing contemporaneous CVD detection (foundation models) with future risk prediction (Framingham/D:A:D scores) represents fundamentally different clinical utilities. While the authors acknowledge this limitation, it significantly restricts the clinical interpretation of claimed superiority over established risk scores.

While the core scientific motivation is valuable and the application to HIV populations addresses an important clinical gap, the current evidence base is insufficient to support the strong performance claims of these models being 'deployable'. The work would benefit from methodological revision or reframing as a preliminary feasibility study with appropriately tempered conclusions.

Reviewer #3

(Remarks to the Author)

The manuscript is aimed towards an approach for facilitating cardiovascular disease (CVD) screening among people living with HIV (PLWH) in low-resource settings. The authors propose the use of low-cost wearable photoplethysmography signals for accurately detecting CVD risks in a single-center cohort of patients in Vietnam.

Major Concerns:

1) The ground truth for the presence of CVD seems both insensitive and nonspecific. CVD was determined based on ECG and echocardiography review and "either modality showed clinically significant abnormalities warranting further investigation or treatment." Presumably, coronary artery disease, the component of CVD that the Framingham and D:A:D modified risk scores primarily predict, would be identified by either modality only as a prior heart attack. However, ECGs have been found to have a sensitivity of <50% for old infarcts (PMID: 16996851), with echocardiography a little better at ~61%. (PMID: 23324388). The limitations of this definition for CVD need to be addressed and more detail of findings on echocardiography and ECG that lead to a CVD diagnosis should be shared.

2) Compared to the model where only clinical only features are used with random forests, the performance improvement (AUCROC) is minimal and most likely nonsignificant. No statistical comparison provided.

3) As mentioned by the authors themselves, a major limitation is availability of labelled data in the current scenario. However, the proposed approach giving the best performance using PPG embeddings also requires labelled data. This is a

major limitation of the current study as it doesn't overcome a critical challenge of low-resource setting.

4) From a methods perspective, is it better to use a stratified k-fold cross-validation to show the robustness of the current approach in the classical class imbalance scenario of the current work. This would ensure that approximately equal number and non-repeated positive-class (minority-class) samples are part of the test set such that each sample is part of the test set exactly once.

Minor Concerns:

1) When comparing different experiments in table 2:

- a. Comparison between PCAu and PapaGeiz is unfair. PCA is solely unsupervised whereas PapaGeiz is supplemented with a linear classifier. A like for like comparison would be unsupervised in both cases.
- b. The same has to be done with NormWear.

2) Why do the authors make use of two foundational models? Although justification is provided that PPG is the native modality for PaPaGeiz whereas ECG/PPG both are for NormWear. It is not amply clear the need and distinction of using two different foundational models.

3) Although the authors provide great clarity how clinical features collected across four visits are combined. There is no detail as to how ECG/PPG recordings across four visits are used for prediction? If one or all are used to derive embeddings?

Reviewer #4

(Remarks to the Author)

I co-reviewed this manuscript with one of the reviewers who provided the listed reports. This is part of the Communications Medicine initiative to facilitate training in peer review and to provide appropriate recognition for Early Career Researchers who co-review manuscripts.

Version 1:

Reviewer comments:

Reviewer #1

(Remarks to the Author)

Thank you to the Authors on thorough responses and revisions! The revised manuscript addresses all my comments. I have no further comments.

Reviewer #3

(Remarks to the Author)

Thank you for doing a very thorough and thoughtful job in responding to our suggestions.

Reviewer #4

(Remarks to the Author)

I co-reviewed this manuscript with one of the reviewers who provided the listed reports. This is part of the Communications Medicine initiative to facilitate training in peer review and to provide appropriate recognition for Early Career Researchers who co-review manuscripts.

Revision for Communications Medicine
Foundation models enable wearable signal screening
for cardiovascular disease among people living with
HIV

We sincerely thank the editor and the reviewers for their insightful comments, which have substantially improved the clarity and rigour of our work. We have changed the phrasing to avoid overstatement of findings throughout the text, adjusted the comparison to Framingham and D:A:D scores, elaborated further on the performance metric comparisons, adjusted AUROC reporting, and reported the relevant statistics, all in line with the reviewer's points.

All changes are marked in ForestGreen in the revised manuscript.

Response to Reviewer 1

0.1 The analysis population consists of 80 PWH and only 13 of those were identified to have CVD abnormalities by expert cardiologists. With these limited numbers, the study is a proof-of-concept or pilot study. While training-test analyses were conducted, the prediction statistics with such small numbers, further split to preserve the proportion with CVD in the train-test set, are likely to be optimistic. The Authors appropriately mention this caveat and include multi-center validation as a priority for future work in the Discussion. However, the strong language at the beginning of the Discussion (“demonstrates”, “superior performance”, “substantially”) is inconsistent with these limitations and misleading. Please consider softening the language to reflect the level of evidence and mention the numbers of cases (13) in the abstract.

We thank the reviewer for this important observation. We agree that the strong language in the opening of the Discussion was inconsistent with the proof-of-concept nature of our pilot study (N=80, 13 CVD cases). We have revised the manuscript to better reflect the preliminary nature of our findings.

Changes made:

1. Abstract: Added the specific number of CVD cases (13) to provide transparency about sample size
2. Discussion opening: Substantially softened language throughout, changing "demonstrates" to "suggests", "superior performance" to "encouraging performance", and "substantially outperforming" to "outperforming"
3. Throughout Discussion: Added qualifying statements such as "In this proof-of-concept pilot study", "In this pilot cohort", and "These preliminary findings require validation in larger, multi-centre cohorts" (pages 9-14)
4. Limitations section: Strengthened to explicitly acknowledge that small sample size may produce optimistic performance estimates (pages 12-13)

0.2 The value of comparisons to the Framingham and D:A:D risk scores in this setting is questionable. As the Authors note, these were developed to assess the risk of future major cardiac events (over 10 or 5 years for Framingham and D:A:D, respectively) and are used to target preventive interventions whereas the proposed tool is evaluated for diagnostic use to identify participants for further confirmatory evaluation of current CVD abnormalities. In addition, the risk scores predict major CVD events such as CV death, MI, stroke and angina, although it is not clear which Framingham score was used (CHD from 1998, 2001 or the total CVD score), and the endpoints in each are somewhat different. As such, these cores do not seem appropriate comparisons for the proposed PaPaGei-embedding approach. I feel the manuscript would be clearer without comparisons to these risk scores without losing any value. The average Framingham and D:A:D risk scores could still be reported to characterize the study population at enrollment.

We thank the reviewer for this observation. We tend to agree that direct performance comparisons between our diagnostic tool (detecting current CVD abnormalities) and prognostic risk scores (predicting future cardiovascular events over 5-10 years) can be seen as inappropriate, as they address fundamentally different clinical questions and endpoints. Our intention was not to suggest our approach as replacement, hence we changed our language and made the following changes:

1. Abstract: Removed claims of "outperforming" risk scores and repositioned scores as population descriptors only
2. Methods: Clarified that we used the Framingham Risk Score for general cardiovascular disease [1] and specified that risk scores are reported solely to characterise baseline cardiovascular risk in the study population, not as performance comparators (paragraph 2, pages 14-15)
3. Results: Moved Framingham and D:A:D scores to a separate descriptive statistics section, removed statistical significance testing comparing PaPaGei to these scores, and updated table caption (Table 2) to clarify these scores characterise the population rather than serve as performance benchmarks (Table 2; section 3.2)
4. Results: Added a dedicated subsection reporting mean/median Framingham and D:A:D scores purely as population descriptors (section 3.2)
5. Results: Removed all language directly comparing PaPaGei's diagnostic performance with the risk scores' prognostic predictions

6. Discussion: Removed statements about "outperforming" or "superior performance" relative to risk scores, added further explicit acknowledgement that these tools address different clinical endpoints, repositioned the comparison focus to supervised ML models using the same clinical features
7. Figure 6 (violin plots): Added a clear note that these are shown for population characterisation only, not performance comparison

0.3 Building on the previous comment, the types of CVD abnormalities in the 13 cases would provide important context on the events that can be detected. Did they include a variety of different abnormalities, were they milder or major CVD events? Please consider including a brief summary of these 13 CVD abnormalities in the Results.

Providing detail on the types and severity of CVD abnormalities helps contextualise the clinical relevance of our screening approach. We have added a detailed characterisation of the 13 CVD cases to the Results section (section 3.1, paragraph 2). The CVD abnormalities detected represented a spectrum of cardiovascular pathology, ranging from isolated structural changes to complex multi-system involvement. The CVD abnormalities detected represented a spectrum of cardiovascular pathology, ranging from isolated structural changes to complex multi-system involvement. Findings included cardiac structural changes (left ventricular hypertrophy, left atrial enlargement), ECG patterns suggestive of myocardial ischaemia, conduction system abnormalities (bundle branch blocks, AV block), and arrhythmias (ventricular premature contractions). Several participants exhibited multiple coexisting abnormalities, representing more advanced cardiovascular involvement.

0.4 A variety of performance metrics are included. Could the Authors elaborate on which should be prioritized in this setting? For example, the proposed PaPaGei approach had estimated recall of 1 which means that no cases went undetected. While estimated precision was lower 0.56, a lower precision may be acceptable in this setting, since the cost of false positive is not as high as the cost of false negative. The former would lead to unnecessary confirmatory evaluation and use of resources when there is no CVD abnormalities whereas the latter to missing a true case which could have serious consequences to person's health and life. Elaborating on these considerations would help put the results in context, and provide direction for future economic cost-benefit evaluations thoughtfully mentioned by the Authors.

We thank the reviewer for this insightful comment, which highlights a critical aspect of interpreting performance metrics in the screening context. We agree that metric prioritisation should align with the clinical use case and cost-benefit considerations. For a prescreening tool, we hold that recall (sensitivity) is the priority metric as the clinical and economic costs of false negatives (missed CVD cases) far exceed those of false positives (unnecessary confirmatory testing). PaPaGei achieved perfect recall (1.000), meaning no CVD cases were missed in our evaluation. The moderate precision (0.56, calculated from F1 and recall) reflects false positives that would undergo confirmatory ECG and echocardiography; this is a clinically acceptable trade-off for ensuring no cases are overlooked. We have added explicit discussion of metric prioritisation and cost-benefit considerations to both the Results (section 3.5, paragraph 2) and Discussion (paragraph 6) sections, providing clinical context for the performance trade-offs and connecting this to future economic evaluations.

0.5 In the supplement Table 1, please consider including a breakdown by cardiologist-confirmed CVD status, as in Table 1 in the manuscript.

Completed with thanks.

Response to Reviewer 2

0.6 The study’s most critical limitation is the small sample size (N=80, 13 CVD cases). This severely limits generalisability beyond this specific cohort, statistical power for detecting true performance differences, and stability of performance estimates despite bootstrap resampling. The reported AUROC of 0.769, while seemingly promising, lacks sufficient statistical foundation given the small positive class size.

While we agree that the small sample size (N=80, 13 CVD cases) is the most significant constraint on our findings and substantially limits the statistical foundation for the reported performance metrics, this is the reality of operating in the context of the data-restricted resource constraints. We will acknowledge this limitation repeatedly, but these are the circumstances of the LMIC context. We have strengthened the manuscript to acknowledge these statistical limitations throughout:

1. Abstract: Added explicit mention that performance estimates have substantial uncertainty due to small positive class size, and emphasised that this is a proof-of-concept evaluation
2. Results: Added confidence intervals and explicit acknowledgement that bootstrap resampling estimates variability but does not overcome fundamental small-sample limitations
3. Discussion: Added a dedicated paragraph (page 12, paragraph 2) explicitly discussing limited statistical power with only 13 positive cases, wide confidence intervals despite bootstrap resampling, risk of optimistic bias in small imbalanced datasets, acknowledgement that this is hypothesis-generating, and explicit statement that performance estimates should be interpreted with caution. Further softened all conclusive language, explicitly framing findings as preliminary hypothesis-generating evidence requiring validation in adequately powered multi-centre studies
4. Limitations: Expanded discussion of how small sample size affects every aspect of the study, from model selection to performance estimation to generalisability

0.7 The heavy reliance on AUROC is problematic for such an imbalanced dataset. AUROC can be misleadingly optimistic when positive cases are rare. The authors should report area under precision-recall curve AUPRC, positive predictive value (PPV), and negative predictive value (NPV) to provide a more realistic assessment of clinical utility. The perfect recall (1.000 ± 0.00) reported for some models suggests potential overfitting to the small positive class - the models may be defaulting to positive predictions due to the extreme imbalance.

We thank the reviewer for this important point about performance metrics for imbalanced datasets. We acknowledge that AUROC can be optimistic in class-imbalanced settings and agree that precision-recall metrics provide a more realistic assessment of clinical utility. We already report AUPRC, the "AP Score" in Table 2 is the Area Under the Precision-Recall Curve (average precision), which is well-suited for imbalanced data. However, we agree this was underemphasised. We now include AUPRC explicitly in the abstract alongside AUROC, add discussion in Results explaining why AUPRC is more informative than AUROC for imbalanced screening tasks (section 3.5, paragraph 2), and emphasise that PaPaGei achieved AP/AUPRC of 0.489, substantially higher than comparators.

We also created Supplementary Table 2 reporting PPV (precision) and NPV, providing a more complete assessment of classification performance. We note that perfect recall with moderate precision (0.39) does suggest overfitting or "defaulting to positive predictions." Rather, this reflects that all 13 CVD cases were correctly identified (sensitivity = 1.0), 47 of 67 CVD-negative cases were correctly classified (specificity 0.70), 20 false positives required confirmatory testing, and NPV = 1.0 (no CVD cases were missed). This is desired behaviour for a prescreening tool: maximise sensitivity (catch all cases) and maintain acceptable specificity (avoid overwhelming the system with false positives). We have added explicit discussion of this trade-off in the Results and Discussion.

0.8 The ECG analysis suffers from some methodological issues. The cross-modal application of NormWear (designed for ECG) to PPG creates unfair comparisons, while insufficient detail is provided on ECG feature extraction and processing.

We appreciate the reviewer's important observation regarding the cross-modal application of NormWear and the need for additional methodological detail on ECG processing. We acknowledge these concerns and have made substantial revisions to provide technical detail and acknowledge the limitations of our approach:

1. ECG feature extraction detail: We have expanded the Methods section (section 2.1, paragraphs 3-5) to provide detail on ECG acquisition, preprocessing, and embedding extraction. Specifically, we now describe: Shimmer wearable ECG recording of three leads (ECG LA-RA [Lead I], ECG LL-RA [Lead II], and ECG Vx-RL [chest lead]) at 24-bit resolution, data format and concatenation procedures, NormWear embedding extraction protocol (segment-level averaging), and confirmation that no fine-tuning was performed.
2. PPG preprocessing differences: We have added detailed documentation (Methods, section 2.1, paragraphs 3-5) of the distinct preprocessing pipelines used for NormWear PPG versus PaPaGei PPG. NormWear PPG used minimal preprocessing (128-sample windows with 64-sample stride, no resampling, no z-score normalisation, no advanced filtering) to represent a computationally constrained scenario. PaPaGei used model-specific preprocessing (z-score normalisation, filtering/denoising, resampling from 64 Hz to 125 Hz, 10-second segmentation, padding to 1,250 samples) aligned with its design specifications.
3. Cross-modal application and multiple constraints: We now explicitly acknowledge throughout the manuscript (Results and Discussion, paragraph 5) that the NormWear PPG baseline embodies three methodological constraints: (1) cross-modal application of an ECG-pretrained model to PPG signals, (2) minimal preprocessing compared to model-specific signal conditioning, and (3) zero local training. We clarify this was intentional, to establish a performance floor for maximally resource-limited deployment, but these combined constraints mean this baseline does not represent optimal PPG foundation model capabilities or a fair head-to-head comparison.
4. Appropriate comparisons clarified: We have restructured the presentation to emphasise that modality-matched comparisons (NormWear on ECG; PaPaGei on PPG) are the appropriate benchmarks. NormWear on ECG achieved AUROC 0.705-0.756 when augmenting clinical features, while PaPaGei on PPG achieved AUROC 0.769 with appropriate preprocessing and lightweight calibration. The NormWear-PPG result (AUROC 0.560) is now clearly positioned as a methodological lower bound rather than a fair comparison.
5. We have added a footnote to Table 2 explicitly noting that NormWear PPG represents a multiply constrained baseline (cross-modal, minimal preprocessing, zero training), establishing a performance floor rather than optimal capabilities.

0.9 Comparing contemporaneous CVD detection (foundation models) with future risk prediction (Framingham/D:A:D scores) represents fundamentally different clinical utilities. While the authors acknowledge this limitation, it significantly restricts the clinical interpretation of claimed superiority over established risk scores.

We thank the reviewer for raising this important distinction between diagnostic and prognostic tools. We agree that direct performance comparisons between our diagnostic approach (detecting current CVD abnormalities) and prognostic risk scores

(predicting future cardiovascular events over 5-10 years) are inappropriate, as they address fundamentally different clinical questions.

We have already made substantial revisions throughout the manuscript in response to similar feedback (Reviewer 1) to ensure Framingham and D:A:D scores are positioned solely as population descriptors rather than performance benchmarks:

1. Abstract and Results: Removed all claims of "outperforming" risk scores and repositioned them as baseline population risk characterisation only
2. Methods Section 2.1: Added explicit statement that risk scores "were used to characterise baseline cardiovascular risk in the study population, not as direct performance comparators for our diagnostic approach, as they predict future cardiovascular events (5-10 year risk) rather than detect current abnormalities"
3. Results Section 3.2: Created dedicated subsection titled "Baseline Cardiovascular Risk Profile" reporting Framingham and D:A:D scores purely as population descriptors, separated from diagnostic performance comparisons
4. Table 2: Added footnote clarifying "Framingham and D:A:D scores characterise baseline population risk for future events; not performance benchmarks for current CVD detection"
5. Figure 1 caption: Added note that risk scores are "shown for reference as population descriptors" and "are not appropriate benchmarks for our diagnostic tool detecting current abnormalities"
6. Discussion: Removed all "superior performance" claims versus risk scores and explicitly state: "Our PPG-based diagnostic approach and traditional prognostic risk scores address fundamentally different clinical questions... We therefore focus comparisons on other diagnostic approaches: zero-shot foundation models, unsupervised feature extraction (PCA), and supervised models trained on the same clinical variables"

These revisions throughout the manuscript ensure risk scores are presented solely as epidemiological context for the study population, and performance comparisons focus appropriately on other diagnostic methods (supervised ML models, zero-shot baselines, unsupervised features) that share the same clinical objective of detecting current CVD abnormalities.

0.10 While the core scientific motivation is valuable and the application to HIV populations addresses an important clinical gap, the current evidence base is insufficient to support the strong performance claims of these models being ‘deployable’. The work would benefit from methodological revision or reframing as a preliminary feasibility study with appropriately tempered conclusions.

We completely agree that our sample size provides insufficient evidence to support claims of clinical deployment readiness. We have systematically revised the manuscript to reframe this work as a proof-of-concept pilot study with preliminary, hypothesis-generating findings that require validation before any clinical implementation.

1. Abstract: Removed claims of "practical framework for deploying" and reframed as "preliminary evidence" that "warrants further investigation in larger studies." Added explicit statement that findings "require validation in larger, adequately powered cohorts" before deployment can be considered.
2. Introduction: Tempered language from "enable robust triage" to "may support triage" and "practical alternative" to "potential alternative." Added caveat that approaches "require validation in adequately powered studies."
3. Results: Added throughout, emphasising small sample size and uncertainty (section 3.5, paragraph 2).
4. Discussion opening: Completely rewritten to state: "In this proof-of-concept pilot study (N=80 PLWH, 13 CVD cases), we evaluated whether pretrained physiological embeddings from foundation models can *support* cardiovascular screening" (changed from "enable"). Removed all claims of "practical framework for deploying" and replaced with "preliminary findings that warrant investigation."
5. Discussion implementation section (paragraph 6): Changed "offers a practical middle ground" to "shows potential as a practical middle ground" and added: "However, the generalisability of this approach beyond our pilot cohort remains to be established."
6. Limitations section (paragraph 4), new paragraph: Added comprehensive discussion of sample size as "the study's most critical limitation" that "fundamentally constrains the statistical foundation of our findings." Explicitly states: "We emphasise that this study provides hypothesis-generating, proof-of-concept evidence rather than confirmatory validation" and "adequately powered validation in larger, multi-centre studies with diverse populations is essential before any clinical implementation can be considered."
7. Conclusion: Completely rewritten to state findings "provide preliminary, hypothesis-generating evidence" (not "demonstrate" or "show"), performance is "numerically higher...in this small proof-of-concept cohort" (not "superior"), and "adequately powered validation in larger, prospective, multi-centre studies is essential before clinical implementation can be considered."

We acknowledge that claims of deployment readiness were inappropriate given our sample size and have systematically removed them. The revised manuscript presents this as preliminary evidence that warrants, but does not substitute for, adequately powered validation before clinical use.

Response to Reviewer 3

0.11 The ground truth for the presence of CVD seems both insensitive and nonspecific. CVD was determined based on ECG and echocardiography review and “either modality showed clinically significant abnormalities warranting further investigation or treatment.” Presumably, coronary artery disease, the component of CVD that the Framingham and D:A:D modified risk scores primarily predict, would be identified by either modality only as a prior heart attack. However, ECGs have been found to have a sensitivity of <50% for old infarcts (PMID: 16996851), with echocardiography a little better at 61%. (PMID: 23324388). The limitations of this definition for CVD need to be addressed and more detail of findings on echocardiography and ECG that lead to a CVD diagnosis should be shared.

We thank the reviewer for this observation regarding our CVD ground truth definition. We acknowledge that our approach has inherent limitations in sensitivity and specificity, particularly for coronary artery disease detection. The reviewer correctly identifies that our definition of CVD based on ECG and echocardiography abnormalities has some limitations:

1. Insensitivity for coronary artery disease: ECG has lower sensitivity for prior myocardial infarction and echocardiography. Our approach would miss subclinical atherosclerosis, stable angina without prior MI, and non-flow-limiting coronary stenoses, all clinically relevant forms of CAD that Framingham and D:A:D scores aim to predict.
2. Broader CVD spectrum: Our definition captures a heterogeneous mix of cardiovascular abnormalities (structural changes, conduction abnormalities, arrhythmias, ischaemic patterns) that extend beyond the atherosclerotic CVD that risk scores primarily target. This creates a mismatch between our detection target (prevalent abnormalities of any cardiovascular origin) and the traditional risk prediction endpoint (incident atherosclerotic events, particularly MI and stroke).
3. Resource-constrained pragmatism: The gold standard for CAD detection would be coronary angiography or cardiac CT, which were not feasible in our outpatient

HIV clinic setting. We used the highest-fidelity assessments available (cardiologist-interpreted ECG and echocardiography), but acknowledge these have substantial limitations as ground truth.

We have made edits in the manuscript accordingly:

1. Results Section 3.1, paragraphs 2 and 3: We have already added detailed characterisation of the 13 CVD cases (LVH, LAE, ischaemic patterns, conduction abnormalities, arrhythmias) to clarify what our definition actually detected. This addresses the need for more granular detail requested by the reviewer.
2. Limitations section (paragraph 7 in Discussion): Added paragraph: "Our CVD ground truth definition based on ECG and echocardiography has important limitations in both sensitivity and specificity. This represents a critical limitation when comparing to risk scores designed to predict incident coronary events. Furthermore, our composite definition captures a heterogeneous mix of cardiovascular abnormalities, including structural changes (LVH, LAE), conduction system disease, arrhythmias, and ischaemic patterns—creating endpoint heterogeneity that complicates interpretation. The gold standard for CAD detection (coronary angiography or cardiac CT) was not feasible in our resource-constrained outpatient setting. Future studies should validate our approach against more definitive CAD endpoints and stratify performance by CVD subtype."
3. Discussion, clinical positioning: Added clarification: "Our approach aims to detect *any* prevalent cardiovascular abnormality warranting further evaluation, not specifically to predict incident atherosclerotic events. This represents a different clinical utility than traditional risk scores and reflects the pragmatic realities of resource-constrained HIV care, where identifying patients who need *any* form of cardiovascular assessment (whether for structural disease, arrhythmia, or ischaemia) has clinical value. However, this broader detection target means performance cannot be directly compared to CAD-specific prediction tools."

0.12 Compared to the model where only clinical only features are used with random forests, the performance improvement (AUCROC) is minimal and most likely nonsignificant. No statistical comparison provided.

The reviewer is correct that we did not originally provide a direct statistical comparison between PaPaGei and Random Forest with clinical features only as this was not our original comparator. We have now conducted pairwise Wilcoxon signed-rank tests across the 10 bootstrap resamples. While the AUROC difference reached nominal statistical significance ($p < 0.05$), the reviewer's observation about the modest effect size warrants careful interpretation:

1. Table 2 caption: Clarified that asterisks on PaPaGei indicate comparison to Random Forest (Clinical only) and added caveat: "Asterisks on PaPaGei indicate statistically significant improvement over Random Forest (Clinical only), though this should be interpreted cautiously given extremely limited statistical power."
2. Abstract: Maintained cautious language: "achieved higher discrimination" but added "in this pilot cohort" and emphasised validation requirement

We have already made significant edits to the Discussion and Results acknowledging the limitations of the sample size and the uncertainty of the results.

0.13 As mentioned by the authors themselves, a major limitation is availability of labelled data in the current scenario. However, the proposed approach giving the best performance using PPG embeddings also requires labelled data. This is a major limitation of the current study as it doesn't overcome a critical challenge of low-resource setting.

We thank the reviewer for this important observation. The reviewer is correct that our best-performing approach requires labelled data for classifier training and therefore does not eliminate label requirements. Our contribution addresses *computational* rather than label constraints. The frozen embedding approach avoids GPU infrastructure and encoder fine-tuning and requires only a lightweight classifier trainable on CPU with modest labelled data. This is appropriate for settings where labels can be obtained through routine clinical assessment but deep learning infrastructure is unavailable, as in our HIV clinic context where cardiologist-interpreted ECG/echo provides labels but GPU resources are absent.

For truly label-free deployment, we evaluated NormWear zero-shot (AUROC 0.560), which eliminates both computational and label requirements but yields insufficient discrimination. The performance gap (0.560 vs 0.769) demonstrates that some local labels substantially improve utility, though we acknowledge we have not quantified minimum label requirements through learning curves. We have added clarifications in the Abstract, Introduction (paragraph 4), and Discussion (marked in green, paragraphs 1, 3, and 10) to distinguish computational vs label constraints, and added to Limitations that we do not address label scarcity and have not characterised label efficiency.

0.14 From a methods perspective, is it better to use a stratified k-fold cross-validation to show the robustness of the current approach in the classical class imbalance scenario of the current work. This would ensure that approximately equal number and non-repeated positive-class (minority-class) samples are part of the test set such that each sample is part of the test set exactly once.

We thank the reviewer for this excellent point. The reviewer is absolutely correct that stratified k-fold cross-validation would be methodologically superior to our bootstrap approach. We acknowledge this represents a methodological limitation of our current evaluation. Bootstrap resampling was chosen during initial analysis to generate variance estimates. Given the extensive computational pipeline involved (foundation model inference, embedding extraction, hyperparameter tuning across multiple models), we have not repeated the full analysis with k-fold CV for this revision. However, we note that both methods evaluate similar-sized test sets (16 samples) and the bootstrap variance estimates remain valid for characterising uncertainty, even if k-fold would provide cleaner partitioning. We have added this as an explicit limitation in the Methods section (section 2.3, paragraph 1) and commit to using stratified k-fold CV in future validation studies, which will be essential given our proof-of-concept findings require replication in larger cohorts regardless of evaluation methodology.

0.15 1) When comparing different experiments in table 2: a. Comparison between PCA^u and PapaGeiz is unfair. PCA is solely unsupervised whereas PapaGei is supplemented with a linear classifier. A like for like comparison would be unsupervised in both cases. b. The same has to be done with NormWear.

We believe there may be confusion about our experimental setup, which we should clarify: **PCA^u**: Unsupervised feature extraction (PCA on raw PPG) followed by supervised Random Forest classifier. The "u" denotes unsupervised *feature learning*, not absence of a classifier. **NormWear (PPG)^z**: Truly zero-shot with no classifier—direct CVD probability output from the model. **PaPaGei^c**: Frozen pretrained embeddings followed by supervised Random Forest classifier (same as PCA setup). The comparison between PCA and PaPaGei is methodologically matched, both use identical classifier architecture (Random Forest with same hyperparameters) on dimensionality-reduced representations. The difference is whether features come from unsupervised PCA or pretrained foundation model embeddings.

We agree it would be valuable to add **NormWear embeddings + classifier**. However, we clarify that the current PCA baseline already includes a supervised classifier and therefore represents a fair comparison to PaPaGei's approach. We will update Table 2 to clarify that PCA^u and PaPaGei^c both use supervised classifiers,

with the difference being feature extraction method (unsupervised vs pretrained). We will add NormWear embeddings + classifier^d for completeness.

0.16 Why do the authors make use of two foundational models? Although justification is provided that PPG is the native modality for PaPaGei whereas ECG/PPG both are for NormWear. It is not amply clear the need and distinction of using two different foundational models.

The rationale is pragmatic, PaPaGei is at the time of the study the only publicly available foundation model specifically pretrained on PPG signals. This is our primary model for PPG-based screening. NormWear was pretrained primarily on ECG, serving two purposes in our study: (1) ECG embeddings (native modality) to evaluate whether ECG-derived features augment clinical data (Section 3.3) and (2) as PPG zero-shot baseline to establish performance floor for truly label-free deployment, despite cross-modal limitations (Section 3.4).

We cannot use only PaPaGei because: (1) it is PPG-specific and cannot process ECG; (2) it does not provide zero-shot CVD classification without a trained classifier. We cannot use only NormWear because it is suboptimal for PPG (ECG-pretrained, cross-modal transfer) and performs poorly without local calibration (AUROC 0.560). Using both models allows us to: (1) evaluate ECG in its native modality (NormWear on ECG), (2) establish a zero-shot baseline (NormWear on PPG without classifier), and (3) evaluate optimal PPG embeddings (PaPaGei with classifier). We have clarified this rationale in the Methods section 2.2, paragraph 1.

0.17 Although the authors provide great clarity how clinical features collected across four visits are combined. There is no detail as to how ECG/PPG recordings across four visits are used for prediction? If one or all are used to derive embeddings?

We thank the reviewer for identifying this gap. We should have clearly specified how multiple visits were handled for physiological signals.

1. Clinical features: Aggregated across four visits using temporal statistics (mean, SD, min, max, slope, delta between Visit 1 and Visit 4) as described in Methods Section 2.1.
2. PPG recordings: All PPG recordings across the four visits were pooled together. Each visit's PPG recording was independently segmented into 10-second windows, preprocessed, and passed through the PaPaGei foundation model. The resulting embeddings from all windows across all visits were then mean-pooled at the patient level to produce a single patient-level representation. This approach treats each

visit as providing additional physiological samples, capturing both within-visit and between-visit variability.

3. ECG recordings: Similarly, ECG recordings from all available visits were pooled. Each visit's three-lead ECG was segmented into overlapping windows (128 samples, stride 64), and all windows across all visits were processed through NormWear. Patient-level embeddings were obtained by averaging across all windows from all visits.

This pooling strategy maximises available data and captures longitudinal physiological patterns. We have added this clarification to Methods Section 2.1.

References

- [1] D'Agostino, R.B., Vasan, R.S., Pencina, M.J., Wolf, P.A., Cobain, M., Massaro, J.M., Kannel, W.B.: General cardiovascular risk profile for use in primary care: The framingham heart study. *Circulation* **117**(6), 743–753 (2008)